# Voltage sensors of a Na⁺ channel dissociate from the pore domain and form inter-channel dimers in the resting state

Ayumi Sumino [1,2] ✉, Takashi Sumikama [1] ✉, Mikihiro Shibata [1,2] & Katsumasa Irie [3] ✉

Understanding voltage-gated sodium ($Na_v$) channels is significant since they generate action potential. $Na_v$ channels consist of a pore domain (PD) and a voltage sensor domain (VSD). All resolved $Na_v$ structures in different gating states have VSDs that tightly interact with PDs; however, it is unclear whether VSDs attach to PDs during gating under physiological conditions. Here, we reconstituted three different voltage-dependent $Na_vAb$, which is cloned from *Arcobacter butzleri*, into a lipid membrane and observed their structural dynamics by high-speed atomic force microscopy on a sub-second timescale in the steady state. Surprisingly, VSDs dissociated from PDs in the mutant in the resting state and further dimerized to form cross-links between channels. This dimerization would occur at a realistic channel density, offering a potential explanation for the facilitation of positive cooperativity of channel activity in the rising phase of the action potential.

Since voltage-gated cation channels (VGCCs) play essential roles in neural transmission, it is important to understand the molecular mechanism of their gating[1]. VGCCs have two structural modules: the voltage sensor domain (VSD) and pore domain (PD). Previous studies have revealed the basic structure of VGCCs; four PDs arranged in a square to form ion pores at the tetrameric center with four VSDs directly attached to the PDs[2–5]. There are two types of arrangements: a VSD on an adjacent subunit associating with a PD (swap type) or a VSD on the same subunit associating with a PD (nonswap type)[6]. In any case, VSDs are tightly associated with PDs. Therefore, it is supposed that voltage-dependent structural changes in VSDs are transmitted to PDs via physical contact and that the formed pore is gated in a voltage-dependent manner[7–11]. The basic structure of these domains and their conformation are similar to those of other voltage-gated Na⁺ channels, K⁺ channels, and Ca²⁺ channels, implying that the molecular arrangement of VSDs and PDs plays a key role in voltage gating[12].

The resting state structure is the least understood structure in the VGCC activation cycle. This is because the formation of this structure requires the resting membrane potential, but it is difficult to analyze

the structure at resting membrane potential. It has only recently been reported that structure of a nonswap-type VGCC under an applied electric field[13]. Regarding the swap-type VGCC, the VSD and PD structures of cross-linked $Na_vAb$ in the resting state have been previously determined[9]. In all the determined structures including those in the activated state, VSDs are believed to be in direct contact with PDs in all structures. However, one exception to this general rule was indicated by a molecular dynamics (MD) study in which MD simulation of the voltage-gated potassium channel predicted that VSDs dissociate from PDs when the channel closes[14]. This implies that there is a possibility of VSDs generally dissociating from the PDs of VGCCs in the resting state. However, this postulated dissociated structure of VSD has not yet been demonstrated experimentally, and thus the question arises as to whether or not this is a computational artifact. If the dissociation event is indeed proved to be real, it would be important to reveal the role of such dissociation in the activation cycle.

All of the resting state structures have been obtained from single-particle analysis[9,13], which allows three-dimensional structures to be determined by averaging a large number of monodisperse protein

¹Nano Life Science Institute (WPI-NanoLSI), Kanazawa University, Kanazawa 920-1192, Japan. ²Institute for Frontier Science Initiative, Kanazawa University, Kanazawa 920-1192, Japan. ³Department of Biophysical chemistry School of Pharmaceutical Science, Wakayama Medical University, Wakayama 640-8156, Japan. ✉e-mail: sumino@staff.kanazawa-u.ac.jp; sumi@staff.kanazawa-u.ac.jp; kirie@wakayama-med.ac.jp

particles observed by cryo-electron microscopy[15]. For collecting and averaging similar structures, cross-linking is sometimes necessary for such structural analysis[9]. If VSDs really move freely away from PDs as found in the MD simulation, then the averaging performed in single-particle analysis is not suitable. In the case of MD simulations, a single individual channel is investigated, albeit via a computational approach. Due to these reasons, further experimental analysis of a single channel without any restriction is needed to completely understand the molecular mechanism of voltage gating.

High-speed atomic force microscopy (HS-AFM) can be used to image the nanostructure and subsecond motion of biological molecules, including membrane proteins[16–22]. HS-AFM can be used to obtain structural information on a single molecule. Therefore, to confirm whether VSDs are always associated with PDs during gating and how Na$_v$ channels interact with each other, we study Na$_v$Ab channels reconstituted in a planar membrane by HS-AFM. We image the nanostructure of the reconstituted Na$_v$Ab channels in different gating states using HS-AFM, which reveals the dissociation of VSDs from PDs in the resting state on the sub-second timescale. Furthermore, dimerized VSDs between channels are observed, and this dimerization is theoretically predicted to be a plausible factor for positive cooperability.

## Results

### HS-AFM of Na$_v$Ab channels in different gating states

We used the Na$_v$Ab channel as a model of the voltage-gated Na$_v$ channel. The Na$_v$Ab channel is the Na$_v$ channel of *Arcobacter butzleri*, which has a relatively simple structure consisting of 268 amino acids per subunit with conserved important parts, such as a PD and VSD, and has been used in many structural studies as a model of the Na$_v$ channel[3,4,9,23]. The Na$_v$Ab channel has six transmembrane helices

denoted as S1 through S6 (Fig. 1a), which form homotetrameric channels. S5 and S6 compose the PD. When the activation gate at the cytoplasmic side of the PD opens, Na$^+$ ions permeate through the pore formed at the center of the tetrameric PDs. Helices S1 to S4 compose the VSD. Between the PD and the VSD, there is an S4-S5 linker. The VSDs of the tetrameric Na$_v$Ab channel surround the PDs by domain swapping (Fig. 1b). On the extracellular side of the VSD, there are two loops linking helices S1 to S2 and S3 to S4 (indicated by the black and white arrows shown in Fig. 1a, b). To reliably observe the single-molecule structural dynamics of the Na$_v$Ab channel on the sub-second timescale by HS-AFM, we truncated the C-terminal cytoplasmic domain (230–268) and added a His-tag on the C-terminus of the PD to suppress diffusion in the bilayer (for readability, this paper will omit describing the notation of the deletion of the C-terminus and the addition of a His-tag). We attached the His-tagged Na$_v$Ab channel onto a Ni$^{2+}$-coated mica substrate, reconstituted it into a lipid bilayer with its extracellular surface facing upward and observed it by HS-AFM (Fig. 1c). To observe the structure of the Na$_v$Ab channel in different gating states in HS-AFM, two mutants, N49K and E32Q/N49K, were used in addition to the wild type (WT). These three constructs have different voltage dependencies; for example, activation starts at −120 mV in the WT, at -50 mV in the N49K mutant, and at 0 mV in the E32Q/N49K mutant, meaning that at 0 mV, the WT channel is fully activated, and the E32Q/N49K channel is mostly in the resting state (Fig. 1d). We also attempted to observe the KAV mutant[9], which is in a complete resting state at 0 mV, but were unable to perform imaging because the tetramer was too unstable and difficult to reconstitute properly in the lipid membrane (Supplementary Fig. 1, Supplementary Movies 1 and 2).

The HS-AFM movies of all Na$_v$Ab channels (WT, N49K, E32Q/N49K) showed tetrameric PDs, whose four particles aligned as squares

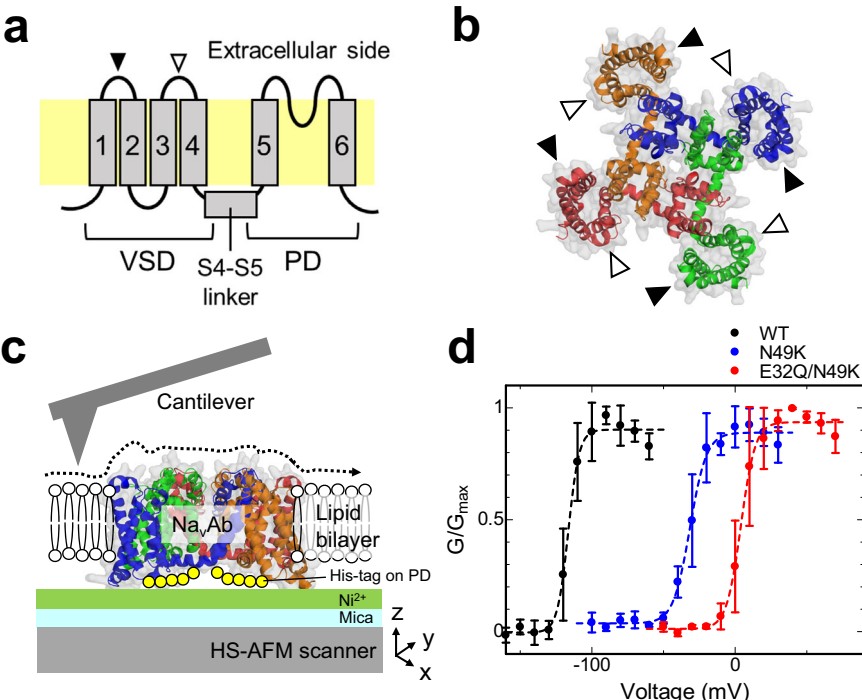

**Fig. 1 | Structure of the Na$_v$Ab channel and HS-AFM sample, and the voltage-dependencies of the Na$_v$Ab constructs. a** Schematic diagram of the Na$_v$Ab channel. **b** Crystallographic structure of the Na$_v$Ab tetramer viewed from extra-cellular side (PDB: 4EKW). The black and white arrowheads in (**a**) and (**b**) indicate the S1-S2 and S3-S4 loops, respectively. **c** Schematic illustration of the HS-AFM imaging of the Na$_v$Ab channel in a lipid bilayer. The Na$_v$Ab channel with a His-tag at the C-terminal is attached to the Ni$^{2+}$-coated mica surface and reconstituted into a

lipid bilayer. The AFM tip was used to scan the extracellular surface of the Na$_v$Ab channel. **d** Normalized conductances of the Na$_v$Ab constructs used in this study. Conductances were normalized by conductances at the maximum tail current of each mutant. Data were obtained from biologically independent cells (WT: $n = 3$, N49K: $n = 4$, and E32Q/N49K: $n = 3$). Symbols and error bars indicate average and standard deviation of normalized conductance, respectively.

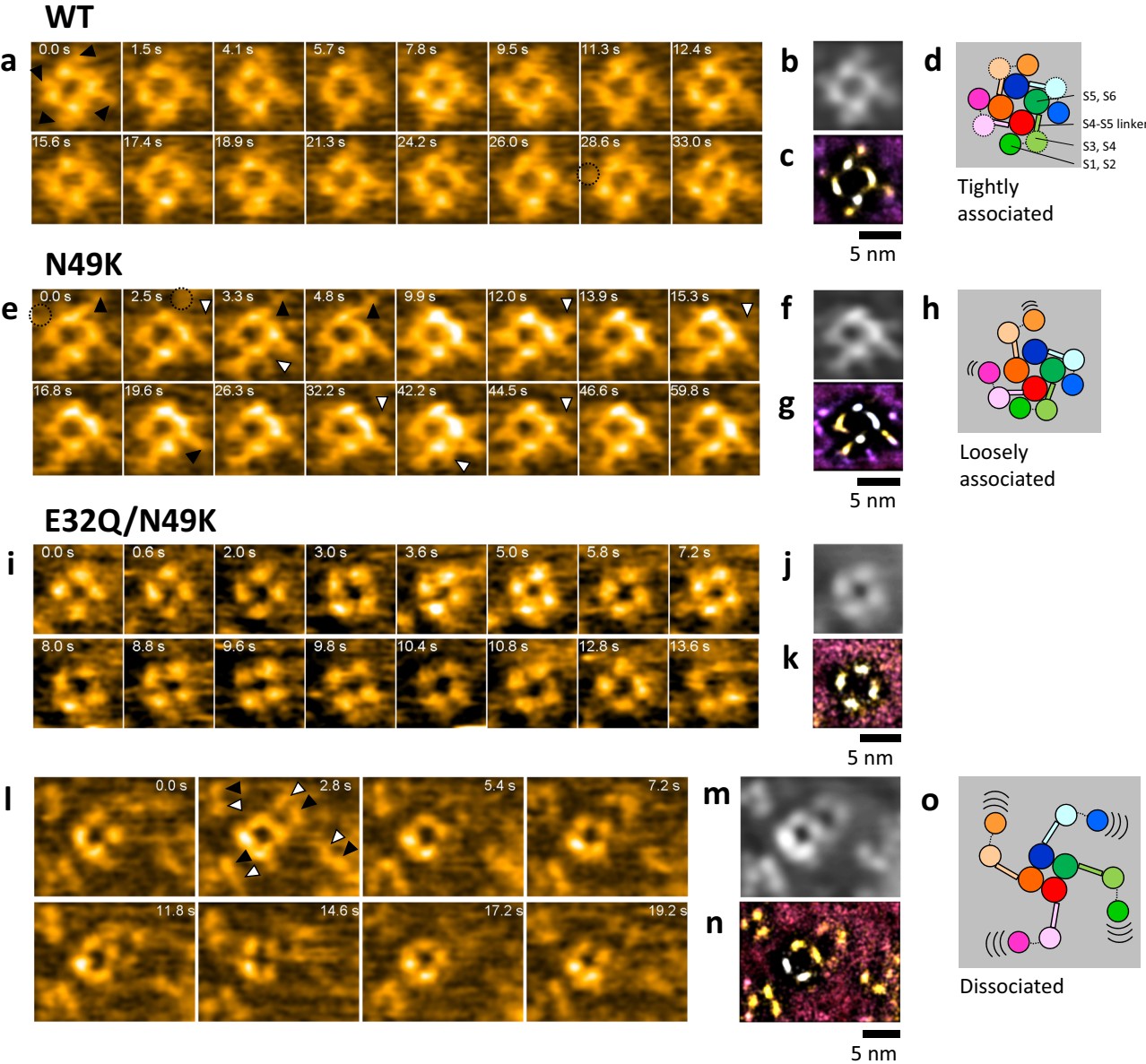

**Fig. 2 | HS-AFM imaging of the Na$_V$Ab channel in a lipid bilayer. a**, **e**, **i**, **l** HS-AFM snapshots of the Na$_V$Ab channel in a lipid bilayer. The black and white arrowheads in (**a**, **e**, **l**) indicate particles derived from the pairs of helices of S1 and S2, and S3 and S4, respectively. Frame rates: 10 frame/s for (**a**, **e**), 5 frame/s for (**i**, **l**). **b**, **f**, **j**, **m** Time-averaged images of HS-AFM movies (**a**, **e**, **i**, **l**), respectively. **c**, **g**, **k**, **n** LAFM images of HS-AFM movies (**a**, **e**, **i**, **l**), respectively. The total frame numbers used for (**b**, **c**), (**f**, **g**), (**j**, **k**), (**m**, **n**) are 458, 599, 65, and 97, respectively. The used constructs are the WT for (**a**–**c**), the N49K mutant for (**e**–**g**) and the E32Q/N49K mutant for (**i**–**n**). **d**, **h**, **o** Schematic illustration of the observed structure of the Na$_V$Ab channels.

(Fig. 2a, e, i, l). This shape is similar to that of the K$^+$ channel without VSDs observed in previous work[24] and is consistent with the fact that both Na$^+$ and K$^+$ channels share the same basic structure of tetrameric PDs.

VSDs were observed around the PDs (Supplementary Movie 3). In the WT channel, four particles surrounded the PDs (black arrows in Fig. 2a). These particles can be seen more clearly in the time-averaged and localization AFM (LAFM)[25] images (Fig. 2b, c). The positions of these particles corresponded to the position of the S1-S2 linkers when compared to the crystal structure (black arrows in Fig. 1b). Thus, the surrounding particles originated from the S1 and S2 of the VSDs (Fig. 2d). Here, particles corresponding to S3-S4 were not seen in the WT channel. In the crystal structure (4EKW)[26], the S3-S4 linker was more embedded in the transmembrane region than the S1-S2 linker, which may explain why no S3-S4-derived particles were observed in the HS-AFM image of the WT channel.

Particles surrounding the tetrameric PDs were also observed for the N49K mutant, which is less likely to open the gate than the WT (Fig. 1d, Supplementary Movie 4). There were two differences observed when comparing this mutant to the WT. One was that some VSDs were not visible in N49K (e.g., upper right in Fig. 2g), while VSDs were almost always associated with PDs in WT. The S1-S2-derived particles at the upper right were clearly observed in frames at 0.0, 3.3, and 4.8 s (black arrowheads) but were barely visible most of the time, suggesting the intermittent binding of the VSDs to the PDs in the N49K mutant. On the other hand, the S1-S2-derived particle in the lower right was observed around the PDs at most moments (e.g., black arrowhead at 19.6 s). This is why the VSDs are not clearly visible relative to the PDs in the time-averaged and LAFM images (Fig. 2f, g). The probability of VSD (S1-S2) detection is summarized in Supplementary Fig. 2. The other difference was that additional small particles (white arrowheads in Fig. 2e) were sometimes observed in a position of S3-S4 (Fig. 1b,

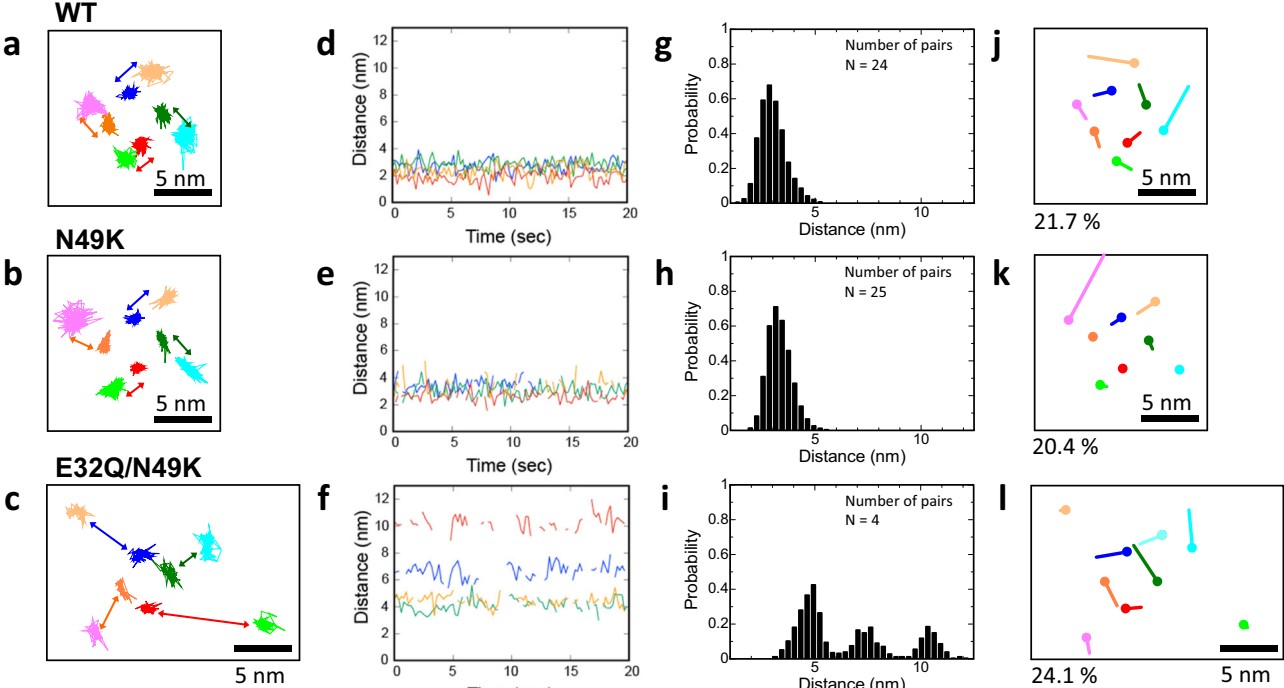

**Fig. 3 | Coupled motions of the PDs and VSDs. a–c** Trajectories of Na$_v$Ab channels. The dark blue, dark green, red, and orange lines correspond to the trajectories of the PDs. The light blue, light green, pink, and light orange lines correspond to the trajectories of the S1-S2 particles. The trajectories of the particles derived from S3-S4 are not shown for simplicity of presentation. **d–f** Time series of the distance between the PD and neighboring VSD (S1 and S2) indicated as colored arrows in (**a–c**). **g–i** Histograms of the distance between the PD and neighboring VSD (S1-S2 particle). The numbers of pairs (channels) are 24 (6), 25 (7) and 4 (1) for g, h, and i, respectively. The total data numbers are 5727, 8231, and 747, respectively. **j–l** Primary components of coupled motion calculated by PCA. The dots and lines indicate eigenvectors of coupled motion, with the dots indicating the origin of vectors. The colors of the vectors are the same as their corresponding trajectories (**a–c**). Below the eigenvectors, the ratio of the eigenvalues of the relevant mode to the sum of the eigenvalues of all modes is shown as a percentage. The analyzed constructs are the WT for (**a, d, g, j**), the N49K mutant for (**b, e, h, k**), and the E32Q/N49K mutant for (**c, f, i, l**).

white arrowhead); it is not clear why these particles were observed in the N49K mutant while they were not visible in the WT. One possible explanation is that some structural change occurred in the VSD, since N49K is a mutation that weakens the interaction between S2 and S4. These observations suggested that the interaction between the VSDs and PDs was weaker in the N49K mutant than in the WT (Fig. 2h).

AFM images of the resting mutant E32Q/N49K were completely different from those of the WT and N49K mutant (Supplementary Movie 5). In the E32Q/N49K mutant, VSDs were not visible in the vicinity of PDs (Fig. 2i, j, k; Supplementary Fig. 2). When the scanned area was enlarged, four two-set particles were observed at a distance away from the PDs (Fig. 2l–n). We concluded that these two-set particles were VSDs for the following reasons. First, no VSDs were observed in the vicinity of the PDs. Second, the distance between the two particles in the set was close to the distance between the S3-S4-derived particle and the S1-S2-derived particle observed occasionally for the N49K mutant (Supplementary Fig. 3). Third, the simulated AFM image of the extracellular surface of a single VSD (4ekw) also showed two particles (Supplementary Fig. 4). The VSDs observed in the E32Q/N49K were clearly visible in the time-averaged and LAFM images, but they were sometimes obscured in the movie (e.g., upper right in snapshot of 17.2 s in Fig. 2l) because they moved around in the lipid bilayer. These were unquestionable data indicating that the VSDs were truly dissociated from the PDs in the resting state of the Na$_v$Ab channel (Fig. 2o).

## Molecular motion of the Na$_v$Ab channels in different gating states

To clarify how the gating states affect the channel motion, we extracted trajectories of each particle from the HS-AFM movies and analyzed the fluctuation of the PD–VSD distance and their coupled motion (Fig. 3). The pair of VSDs and PDs were assigned by taking the swapped structure into account (Fig. 2d, h, o), and the PD–VSD distances were estimated. The distance between center of PD particle and center of VSD particle was defined as the PD–VSD distance. In the E32Q/N49K mutant, we assumed the closest VSD in the clockwise direction from the PD as the VSD linked to the PD. From the trajectories, we calculated the time variation of the distance from each VSD to the PD (Fig. 3d–f). According to the histograms analyzed from the distances between all pairs, the distances in the N49K mutant and WT were similar (Fig. 3g, h). In contrast, although the E32Q/N49K mutant showed a multipeaked distribution because of an insufficient sample number (Fig. 3i), even the nearest peak of the E32Q/N49K mutant was much farther away than those of the WT and N49K mutant. Using the measured distances and two assumptions, the maximum distance between the PD and S1-S2 was found to be approximately 7.5 nm (Supplementary Fig. 5, and see explanations in the caption). Although it was not possible to determine from the images which particles were connected to each other intracellularly since the extracellular side was observed, it is possible that the S4-S5 linker was unfolded and disordered because the observed distance between the PDs and S1-S2 particles shown in Fig. 3i (3–11 nm) was much longer than the estimated maximum distance (7.6 nm).

Principal component analysis (PCA) was performed to investigate how the motions of the VSDs were coupled to those of the PDs. PCA can visualize the so-called collective mode. In this analysis, eigenvectors appeared if the motions were strongly coupled; otherwise, no vectors appeared. In the WT channels, vectors appeared along the rotational direction for most of the PDs and VSDs (Fig. 3j and

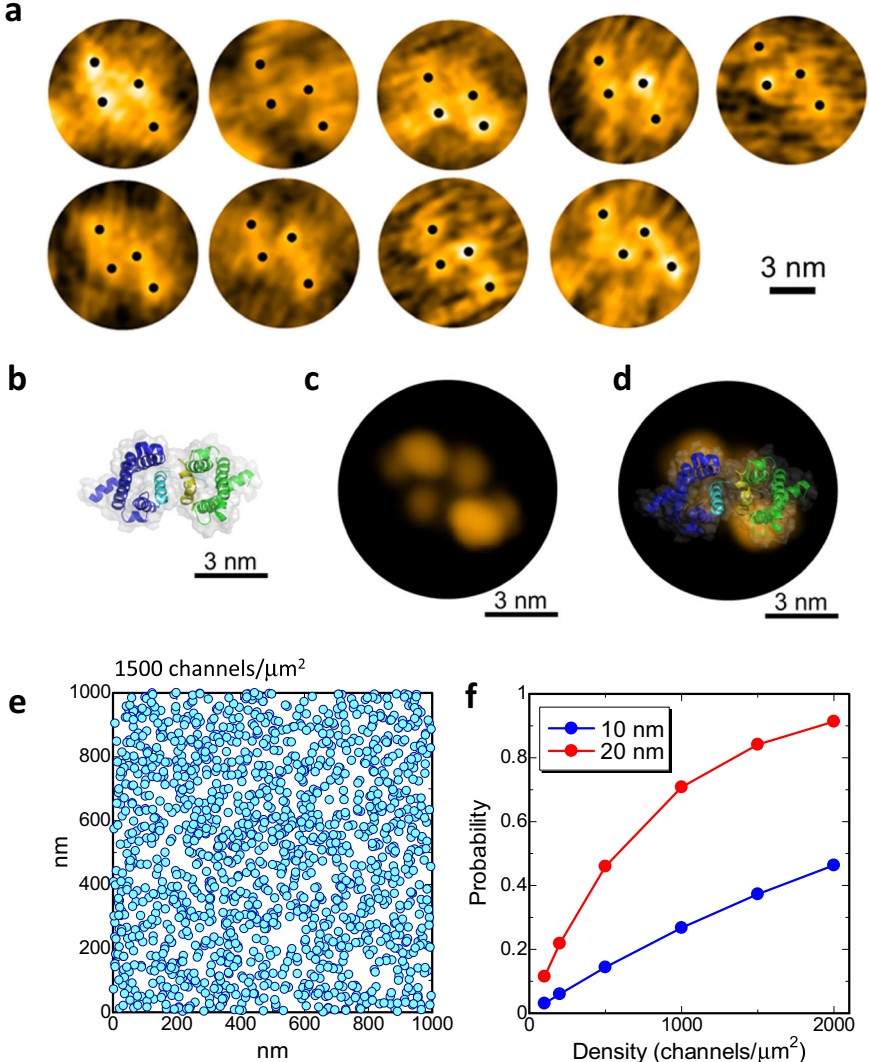

**Fig. 4 | Inter-channel dimerization of VSDs. a** HS-AFM snapshots of sets of four particles arranged in parallelogram. All panels show different sets of four particles. The black dots indicate the centers of particles. The molecules are aligned at a similar angle. **b** Structure of a VSD dimer after 1 μs MD simulation. The S4 helices are colored in cyan and yellow. **c** Averaged-simulated AFM image of the extracellular surface of the modeled dimer. The virtual membrane surfaces were set at 4 nm height from the intracellular surface, and the assumed tip radius was 0.5 nm for the simulated AFM image. The time-averaged image was calculated by averaging 100 simulated AFM images, in which they were calculated by using 100 structures extracted from 1 μs MD simulation. **d** Overlay image of (**b**, **c**). Note that the molecular orientation in (**b**) is the same as that in (**c**). **e** Particles of 20 nm diameter placed randomly in an area of 1 μm². The diameter roughly corresponds to that of the resting $Na_vAb$ channel. The center coordinates of the particles were randomly generated without considering the exclusive area of other particles, so the particles were allowed to overlap each other. The density is 1500 channels/μm². **f** Contact probability at different densities of particles with diameters of 10 and 20 nm. Probabilities were calculated using 10000 sets of randomly generated snapshots, including 100–2000 particles in area of 1 μm². The proportion of particles in contact with at least one other particle was plotted.

Supplementary Fig. 6). This indicated that the rotational motions of PDs were tightly coupled to those of VSDs, which is expected since they were seen in the vicinity in the HS-AFM images. On the other hand, for the N49K mutant, a few large vectors appeared for the VSDs, but almost no vectors appeared for the PDs (Fig. 3k and Supplementary Fig. 7). This was also the case for the second and third principal components (Supplementary Fig. 7). This indicated that the motions of VSDs were not coupled to those of PDs and supported the intermittent binding of VSDs to PDs observed in the AFM images. Accordingly, it strongly suggested that the interaction between VSDs and PDs was almost lost in the N49K mutant, even though the VSDs were observed to be in the vicinity of the PDs in the HS-AFM images. For the E32Q/N49K mutant, VSDs were observed to be dissociated from PDs, resulting in a minor degree of coupling (Fig. 3l and Supplementary Fig. 8).

## Dimerization of dissociated VSDs in the resting state
In the reconstituted membrane of the E32Q/N49K mutant, there were sets of four particles arranged not in a square but in a parallelogram (Fig. 4a). This structure can also be seen clearly in the lower lefthand corner of Fig. 2n. The dissociated VSDs showed two particles derived from S1-S2 and S3-S4 (Fig. 2l–n); thus, we assumed that these sets of four particles were arranged in parallelograms as dimers of two VSDs. To confirm this clearly, the equilibrium structure of the VSD dimer in the membrane was obtained by MD simulation using a homology model based on a dimer of the $H_v$ channel as the initial channel structure (Fig. 4b). The $H_v$ channel is a proton channel composed solely of VSD-like domains that form a dimer and cooperatively gates[27–31]. In the simulated structure, S4 helices interacted to form the channel interface in the half of the membrane facing the intracellular solution (Supplementary Movie 6). The S4 and S3 helices interacted

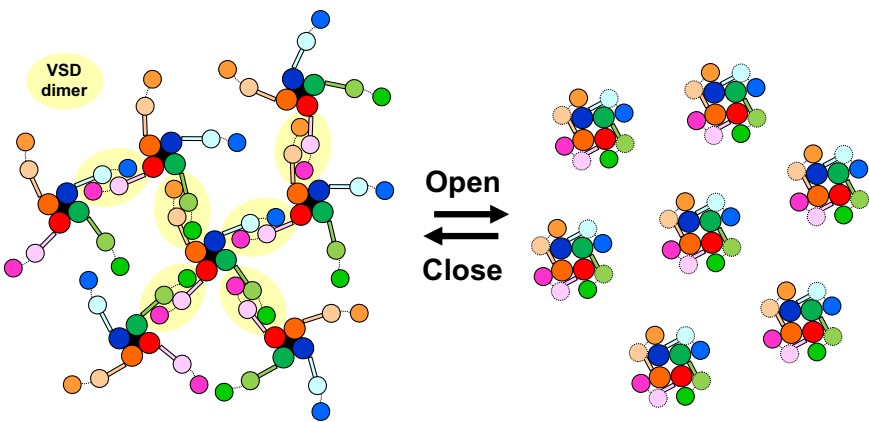

**Fig. 5 | Inter-channel network between resting channels and cooperative gating.** Suggested structural changes associated with the voltage-gating of Na$_v$ channels; resting Na$_v$ channels form a network via VSD dimers, and the VSD dimers dissociate and attach to the PDs upon activation.

with the PDs in the activated conformation[3], meaning that this dimeric structure was formed only when the VSDs were detached from the PDs. Using the MD simulation data, we computed the time-averaged image of the simulated AFM images[32] of the VSD dimer (Fig. 4c, d). The simulated AFM images showed four particles arranged in a parallelogram similar to the experimental AFM images, suggesting that the observed structures were dimers of VSDs.

**Estimation of dimerization probability in the cell membrane**

To estimate whether the channels could interact with each other with a realistic density of channels in the cell membrane, we calculated the contact probability with densities in the range of 100–2000 µm². The density of the Na$_v$ channels at the Ranvier node is -1500 channels/µm²[21,33,34]. The coordinates of particles were randomly generated, and contact probabilities were calculated with different particle sizes and densities (Fig. 4e, f and Supplementary Fig. 9). Note that we generated coordinates without considering the exclusive area occupied by other particles to estimate the lowest contact probability; the actual contact probability should be slightly higher than this estimation. Assuming a particle diameter of 20 nm, which was the size of the resting channel seen in Fig. 2l, 84% of channels could interact with at least one other channel at a density of 1500 channels/µm² (Fig. 4f). Therefore, it is plausible that when VSDs dissociate from PDs, VSDs can cross-link between at least one other channel in the node of Ranvier. If we assume the size of the Na$_v$ channel to be 10 nm (10 nm roughly corresponds to the diameter of the activating channel), the contact probability becomes 37% (Fig. 4f). This simple estimation suggests that, in addition to the well-known structural changes of individual VSDs, collective structural changes, such as the formation and dissociation of inter-channel networks, occur when the activation gate of the Na$_v$ channel opens and closes (Fig. 5).

## Discussion

**Dissociation of VSDs from PDs in the resting state**

The VSDs have been considered to be tightly bound to the PDs in the WT so far. This tight-bound structure corresponds to a fully activated structure in the depolarization state in this study. In the E32Q/N49K mutant, in which some VSDs were not fully activated without a membrane potential, all VSDs were found to dissociate from the PDs. The N49K mutant, whose voltage dependency is between the WT and E32Q/N49K, appeared to have VSDs that were tightly attached to the PDs, as in the WT, but the PCA revealed that the interaction between VSDs and PDs was diminished (Fig. 3j–l). Elinder et al. reported that the distance between the extracellular tops of the S4 and S5 helices was 8 Å in the open state, but was elongated > 20 Å in the closed state[35]. The present observations are consistent with this result. During hundreds of

µs MD simulation of the K$_v$ channel, a channel of the VGCC family, it was observed that the S4-S5 linker unraveled and the VSDs detached from the PDs and moved far away[14]. Our results offer experimental evidence for this MD simulation result. Even in the N49K crystal structure, VSDs have been observed to be slightly farther from the PDs compared to those in the WT crystal structure[4]. It is expected that the tight binding of VSDs to PDs is strengthened by structural changes in the VSDs associated with the depolarization of the membrane potential. Accordingly, a consideration of the association and dissociation of VSDs from PDs would be beneficial for elucidating the gating mechanism of VGCCs.

The single-particle structure determined for the resting state of the swap-type VGCC was derived from a mutant channel in which two residues were replaced by cysteine to form a disulfide bond between VSD and PD[9]. Without the disulfide bond, the VSD structure of this channel formed the activated state[9], and has not therefore been determinable by single-particle analysis. Therefore, the VGCC structure in the resting state without the disulfide bond has not been reported. This implies that the known structure in the resting state was one of the snapshots fixed by the disulfide bond and that VSDs may undergo more significant movement when VGCCs are functionally expressed without the disulfide bond in the resting state. Furthermore, even for the resting state channel structure determined with the disulfide bond present, the cytosolic-side conformation of the S4 helix differed from that of the activated state because the extracellular side of VSDs and PDs were fixed by the disulfide bond[9]. Therefore, under natural conditions without the disulfide bond, the possibility is extant that the cytosolic-side of the S4 helix has other structures, such as the elongated S4-S5 linker.

In this study, we performed HS-AFM on Na$_v$Ab channels. The use of HS-AFM may prove to be the key to observing the VSDs dissociation from PDs in the resting state. Structures of dissociated VSDs have not been observed thus far because it is difficult to detect them with single-particle-based structural analysis, which requires averaging a great large number of molecular structures[9,36]. Even if VSD dissociates in a natural membrane, it will be difficult to classify and determine the dissociated structure of VSD by image classification and averaging of single particle analysis, because the structures are various (Fig. 2). Also, various structures might be consolidated into a single conformation in the process of crystal growth. In stark contrast to such considerations, HS-AFM is a method that can obtain structural information from a single molecule, which allowed us to observe the dissociated VSD structure and monitor fluctuations on a sub-second scale in the steady state.

VGCCs include a swap-type VGCC in which the VSD contacts the PD of the adjacent subunit and a nonswap-type VGCC in which the VSD

contacts the PD of the same subunit[6]. Comparing these two types, we note that the position of the VSD is very different. Even in swap-type VGCCs, there are various VSD positions[37]. Thus, there is significant diversity in the VSD positions, which implies that the VSD positions have changed during the molecular evolution of VGCCs. It suggests that VSD could significantly move around the PD. In a recent study, the VSD arrangement of $K_vAP$ channel, the first VGCC whose structure was determined[38], was finally confirmed to be a nonswap type after a period of controversy[39]. This further suggests that it is difficult to determine the actual position of VSD, and it is necessary to analyze VSD arrangement with various methods. Therefore, it will be interesting to obtain more information about the possible dissociation of VSD using HS-AFM.

### Possible gating mechanism due to VSD dissociation in the resting state

The phenomenon that the VSD is distant from the PD in the resting state does not align with the model of the gating mechanism in which the S4-S5 linker helix closes the channel due to the sinking of the S4 helix in the resting state, and the S4-S5 linker helix loosens because of the rising of the S4-S5 linker helix upon depolarization and opens the channel[9]. The fact that dissociated VSDs cause PDs to close raises the question of what causes the closing of the channel. Lipids can be considered as candidates to study how closing occurs without VSDs. That is, when the VSD is dissociated from the PD, lipids associate tightly with the PD. Recently, the single-particle analysis of large pore channels has suggested that lipids may be directly involved in channel gating[40–42]. In these cases, it is predicted that lipids directly fill the pores of the channels. There may be a mechanism by which tightly adhering lipids cause PDs to close the channel, and then activated VSDs approach the PDs and replace the lipids around the PDs to reopen the channel. Again, note that our observation aligns with the gating mechanism observed in the MD simulation, wherein the VSD dissociates when the channel closes[14]. Detailed analyses of such MD simulation trajectories are necessary to draw conclusions.

### Dimerization of dissociated VSDs

While the inter-channel dimerization of dissociated VSDs in the resting state of the $Na_v$ channel is unreported, the observed dimer structure is similar to that in the structural model of an $H_v$ dimer[29] that is an $H^+$ channel composed of only a VSD-like domain[27–31]. It has also been reported that VSDs interact with other VSDs during early S4 transitions[43]. While it was assumed that the coupled activation between VSDs occurred in a single channel, this observation could also be considered an inter-channel interaction based on our results. Voltage-gated phosphatase, another example of a protein with a VSD, has also been reported to be a dimer[44]. S4 is usually attached to PDs in the activated state. Thus, it is possible that the VSD itself has a dimerization ability, and dimerization is a common phenomenon in the VGCC family.

In the case of VGCCs, it has not been assumed that there is interaction between VSDs within a channel because of the steric hindrance of PDs since VSDs were considered to be in direct contact with PDs. In our previous study, the gradual inactivation of the channel current was observed in NaChBac, the VSD of which had a single cysteine mutation[45]. This inactivation occurred when the concatemer of the NaChBac tetramer was mutated by the introduction of cysteine residues into not adjacent VSDs but into diagonal VSDs. The HS-AFM results provide a clear answer to the gradual inactivation of a single cysteine mutant. By applying a deep negative potential for the first time during whole-cell patch clamp measurement, NaChBac enters a resting state, causing VSD to dissociate from PD and cross-link with VSD on other channels. When VSD cross links with that of other channels, it is possible that the channel is gradually deactivated because the channel cannot be activated.

Recently, Clatot et al. reported that $Na_v1.5$ forms a dimer by interacting with a 14-3-3 protein[46,47] and shows coupled gating[48]. By combining experimental data[47,48] and the structures of $Na_v1.5$[2] and the 14-3-3 protein[46], the dimeric structure of $Na_v1.5$ has been modeled[49]. In this model, two VSDs from different channels associate with each other with the supportive binding of the 14-3-3 protein. We propose another possibility: the VSDs of Nav1.5 form a dimer prior to the binding of the 14-3-3 protein, and then the 14-3-3 protein binds and stabilizes it. To our knowledge, there are no conflicting reports of the dimerization of dissociated VSDs.

### Implication for clustering and functional cooperativity

Historically, the Hodgkin-Huxley equations have been used to model action potential generation[50,51]. Naundorf et al. have proposed that the cooperative activation of sodium channels is important for reproducing the experimental dynamics of action potential initiation[52], though the molecular basis of functional cooperativity is unclear. There is still a controversy over the cooperativity[53,54], while the functional cooperativity of $Na^+$ channels has been extensively investigated[49,52,55–62]. Since it has been reported that dimers of $H_v$ channels are cooperatively activated[30], the dimerization of VSDs might be involved in the cooperative activation of $Na_v$ channels. $Na_v$ channels are densely distributed, especially at the node of Ranvier in myelinated nerves[63,64], with a density of ~1500 channels/$\mu m^2$[21,33,34]. Recent super-resolution microscopic imaging of the node of Ranvier showed the nanoscale localization of $Na_v$ channels with 190-nm periodicity in the node[64], meaning that the local density around $Na_v$ channels is much higher than 1500 channels/$\mu m^2$. Even though the channels are randomly arranged at a density of 1500 channels/$\mu m^2$, 84% of channels can interact with other channels when their diameter is 20 nm, which corresponds to the diameter of the resting channel observed in this study (Fig. 4f). Thus, in the actual node of Ranvier, it is plausible that a network of inter-channel cross-linking via dissociated VSDs is formed. Interestingly, Huan et al. reported that a small fraction, 5–15%, of strongly cooperative channels generate action potentials with the most rapid onset dynamics[57]. Based on our simple estimation, ~12% of channels with a 20 nm diameter interact with four other channels at a density of 1500 channels/$\mu m^2$ (Supplementary Fig. 9i), implying that these channels are strongly cooperative channels. Integrating all this information, we expect that the dimerization of dissociated VSDs stabilizes the resting state of channels and is a candidate for the molecular basis of the cooperative activation of $Na_v$ channels (Fig. 5).

Many other voltage-gated $K^+$ and $Ca^{2+}$ channels in the VGCC family also form clusters and cooperatively gate[49,65–69]. In addition to the VGCC family, the KcsA channel, sharing the structure of PDs, showed clustering-dispersion upon gating[70]. Therefore, abnormality in inter-channel association in the VGCC family, such as mutations in the residues between VSDs, would be the cause of some channelopathies. Since the basic structure of PDs and VSDs in voltage-gated channels is shared among VGCCs[1], it would be very interesting if the inter-channel cross-linking by dissociated VSDs in the resting state is widely shared in other voltage-gated channels and is the molecular basis for their cooperative activation.

## Methods

### Expression and purification of $Na_vAb$

The $Na_vAb$ mutated DNAs were subcloned into the modified pBiEX-1 vector (Novagen) that was modified by replacing the fragment from NcoI site (CCATGG) to SalI site (GTCGAC) in multicloning site with the sequence "CCATGGGCAGCAGCCATCATCATCATCATCACAGCAGCGG CCTGGTGCCGCGCGGCAGCCATATGCTCGAGCTGGTGCCGCGCGGC AGCGGATCCTAAGTCGAC."[4] To add the N-terminal hexa-histidine tag and thrombin cleavage site for protein purification, the NavAb mutated DNAs were subcloned between BamHI and SalI site. The polymerase chain reaction accomplished site-directed mutagenesis using

PrimeSTAR® Max DNA Polymerase (Takara bio; R045A.). For AFM measurement, all Na$_v$Ab channels were truncated at the C-terminal 37 amino acids from His231 to C-terminal Asn267, corresponding to the cytosolic helix, and a 7 amino acid linker (ENLYFQG) and C-terminal hexa-histidine tag without a thrombin digestion site were added. For brevity, the notation of the deletion of the C-terminus and the addition of the His-tag were omitted in the main text. All clones were confirmed by DNA sequencing.

Proteins were expressed in the *Escherichia coli* KRX strain (Promega; L3002). Cells were grown at 37 °C to an OD$_{600}$ of 0.6, induced with 0.2% α-L (+)-Rhamnose Monohydrate (Fujifilm Wako; 182-00751), and grown for 16 h at 25 °C. The cells were suspended in TBS buffer (20 mM Tris-HCl pH 8.0, 150 mM NaCl) and lysed using a high-pressure homogenizer LAB1000 (SMT Co. Ltd.) at 12,000 psi. Cell debris was removed by low-speed centrifugation (12,000 × $g$, 30 min, 4 °C). Membranes were collected by centrifugation (100,000 × $g$, 1 h, 4 °C) and solubilized by homogenization in TBS buffer containing 30 mM n-dodecyl-β-D-maltoside (DDM, Anatrace; D310). After centrifugation (40,000 × $g$, 30 min, 4 °C), the supernatant was loaded onto a HIS-Select® Cobalt Affinity Gel column (Sigma-Aldrich; H8162-100ML). The protein bound to the cobalt affinity column was washed with 10 mM imidazole in TBS buffer containing 0.05% lauryl maltose neopentyl glycol (LMNG, Anatrace; NG310) instead of DDM. After washing, the protein was eluted with 300 mM imidazole, and the N-terminal His-tag was completely removed by 5U/ml thrombin (Fujifilm Wako; 206-18411) digestion in eluted solution added 2 mM CaCl$_2$ (overnight, 4 °C). This digestion condition was established in our previous study[4]. Eluted and N-terminal His-tag removed protein was purified on a Superdex-200 column (Cytiva; 28990944) in TBS buffer containing 0.05% LMNG. Excessive surfactants were removed from the gel-filtration fraction of Na$_v$Ab by GraDeR[71]. The top layer buffer contained 20 mM Tris (pH 8.0), 150 mM NaCl, 5% glycerol, and 0.003% LMNG. The bottom layer buffer contained 20 mM Tris (pH 8.0), 150 mM NaCl, and 25% glycerol. After stacking the buffers, the gradient was generated with Gradient Master 108 (BioComp Instruments). The gel-filtration fractions containing the Na$_v$Ab proteins were loaded on top of the gradient and centrifuged at 35,000 rpm (209,678 g) for 18 h at 4 °C with an SW41Ti rotor. The solution was fractionated by a peristaltic pump from bottom to top. The fractions containing Na$_v$Ab protein were detected by tryptophan fluorescence-detection size exclusion chromatography. The protein fractions were dialyzed against 20 mM Tris (pH 8.0) and 150 mM NaCl buffer overnight at 4 °C and concentrated to a protein concentration of 0.1 mg/ml.

## Electrophysiological measurement of the Na$_v$Ab

The recordings were performed using SF-9 cells. SF-9 cells (ATCC catalog number CRL-1711) were grown in Sf-900™ II medium (Gibco; 10902096) complemented with 0.5% 100× Antibiotic-Antimycotic (Gibco; 15240062) at 27 °C. Cells were transfected with target channel-cloned pBiEX vectors and enhanced green fluorescent protein (EGFP)-cloned pBiEX vectors using FuGENE HD transfection reagent (Promega; E2311). First, the channel-cloned vector (1.5 μg) was mixed with 0.5 μg of the EGFP-cloned vector in 100 μL of the culture medium. Next, 3 μL of FuGENE HD reagent was added, and the mixture was incubated for 10 min before the transfection mixture was gently dropped onto cultured cells. After incubation for 16–48 h, the cells were used for electrophysiological measurements.

For measurement, a pipette solution contained 135 mM NaF, 15 mM NaCl, 10 mM EGTA, and 10 mM HEPES (pH 7.4 adjusted by NaOH), and a bath solution contained 150 mM NaCl, 1.5 mM CaCl$_2$, 1.0 mM MgCl$_2$, 10 mM HEPES (pH 7.4 adjusted by NaOH) and 10 mM glucose. Cancellation of the capacitance transients and leak subtraction were performed using a programmed P/10 protocol delivered at −140 mV. The bath solution was changed using the Dynaflow® Resolve system. All experiments were conducted at 25 ± 2 °C using a whole cell

patch clamp recording mode with a HEKA EPC 10 amplifier and Patch master data acquisition software (v2x73). Data export was done using IGOR 6.37 and NeuroMatic (version 3.0b)[72]. All sample numbers represent the number of individual cells used for each measurement. Cells that had a leak current smaller than 0.5 nA were used for measurement. When any outliers were encountered, these outliers were excluded if any abnormalities were found in other measurement environments and were included if no abnormalities were found. All results are presented as the mean ± standard error.

## Preparation of destabilized liposomes

We dissolved the phospholipid 1,2-dimyristoyl-sn-glycero-3-phosphocholine (DMPC) in chloroform in a glass tube and dried for at least 2 h in vacuo. The lipid film at the glass bottom was hydrated by buffer (10 mM HEPES [pH 7.5], 200 mM KCl) to form a liposome solution (10 mg/mL). n-Dodecyl-β-D-maltopyranoside (DDM) was added to the liposome solution at 0.06% to solubilize the liposomes. This comicellar solution of DDM and DMPC was diluted with DDM-free buffer (20 mM Tris, [pH 7.4], 135 mM NaCl, 20 mM KCl) to a final DDM concentration of 0.006%, resulting in DDM-destabilized liposomes. All procedures were performed at room temperature.

## Reconstitution of Na$_v$Ab into lipid bilayers

We reconstituted the Na$_v$Ab channel into a lipid bilayer by a previously reported method[73,74]. To suppress diffusion in the bilayer, channels were immobilized on the mica surface through a Ni$^{2+}$–His-tag interaction. First, solubilized Na$_v$Ab channels with His-tag at cytoplasmic C-termini (1–10 μg/mL in 20 mM Tris, [pH 7.4], 150 mM NaCl) were applied onto Ni$^{2+}$-coated mica surface for 5 min, then excess channels floating in solution were washed out by buffer (20 mM Tris, [pH 7.4], 150 mM NaCl). Next, DDM-destabilized DMPC liposomes (50–200 μg/mL DMPC) were applied, incubated for 5–10 min, and then rinsed with buffer (20 mM Tris, [pH 7.4], 150 mM NaCl). All procedures were performed at 25 °C. Since the His-tag was on the cytoplasmic side, the extracellular side of the Na$_v$Ab channel faced upward in the resulting reconstituted membrane and was imaged by HS-AFM.

## HS-AFM observation and image processing of reconstituted Na$_v$Ab

We used a laboratory-built HS-AFM[16,75] and the cantilever AC7 (Olympus Co., Tokyo, Japan) with an electron beam deposition tip. The shape of the tip was conical, and the tip radius was less than 2 nm based on the obtained resolution. The typical resonant frequency and quality factor in water of AC7 are 700 kHz and 2, respectively. Imaging buffers were 20 mM Tris, (pH 7.4), 150 mM NaCl for WT and E32Q/N49K, and 50 mM NaCl for N49K. The data from the 50 mM NaCl condition were used in the analysis because the resolution was higher in N49K, but the overview of the channel structure did not change at either 50 mM or 150 mM. Scanning rates were 5 or 10 frames/s (the actual rate is described in the figure caption), measuring fluctuations on a sub-second scale in the steady state. Typical scanning range was 50 nm × 30 nm. These scanning conditions were optimized to obtain high-resolution images without disrupting the tetrameric structure of channels. Any faster scanning rates or larger scanning areas would destroy the structure of the sample. Time-averaged images and LAFM images are calculated by the z-project and LAFM plugin[25] on ImageJ after removing noise and tilt as described in the next section. To compensate for changes in molecular size due to piezo nonlinearity and other factors, the movies were scaled so that the average distance between adjacent PDs was 3.3 nm. We used BioAFMViewer[32] for the simulation of AFM images.

## Particle tracking in HS-AFM movies

To track particles correctly, we removed background, high-frequency noise and lateral drift of HS-AFM movies using subtracting

**Table 1 | The setup information of the simulation**

| VSD monomer | 2 |
|---|---|
| POPC lipid molecule | 249 |
| Water molecule | 21,206 |
| Na$^+$ ion | 58 |
| Cl$^-$ ion | 68 |
| Total number of atoms | 101,040 |
| Ion concentration | 150 mM |
| Simulation box dimensions | 113.394 × 86.271 × 100.178 Å$^3$ |

background, FFT filter and template matching plugin on ImageJ, respectively. We detected and tracked particles by using the Track-Mate plugin[76] on ImageJ. The estimated particle size was set to 2.5 nm, which corresponds to a longer diameter of two α-helices when they are close together. Obvious particle misrecognitions and trajectory swaps were corrected manually. Only for the PCA described below were coordinates generated by linear interpolation for frames where no particles were recognized. In E32Q/N49K, we assigned the two VSD-derived particles as follows: the closest particle clockwise from the PD with the center of the tetramer as the origin as S3-S4 particles and the next closest particles as S1-S2 particles.

## PCA
PCA was applied using the conventional method[77], that is, by diagonalizing the covariance matrix ($C$) defined as follows:

$$C_{ij} = \left\langle \left( \vec{r}_i - \langle \vec{r}_i \rangle \right) \left( \vec{r}_j - \langle \vec{r}_j \rangle \right) \right\rangle,$$

where $\vec{r}$ represents the x and y positions of the center pixel of particles detected by TrackMate in ImageJ and angular brackets represent the time average. Alignment of the frames was not performed, since the drift of PDs was small. In this analysis, four-fold symmetry was not taken into account to consider asymmetric motion of VSDs and PDs. In other words, each particle was considered to move independently. By diagonalizing $C$, eigenvectors and eigenvalues were computed.

## Modeling of VSD dimer and MD simulation
In the homology modeling, the H$_v$ dimer was obtained by super-imposing the H$_v$ monomer (pdb code: 3wkv) on each helix of the cytosolic coiled-coil helices structure of H$_v$ (pdb code: 3vmx). The VSD of Na$_v$Ab was extracted from the Na$_v$Ab crystal structure (PDB code: 5yuc) and superimposed on the Hv dimer based on the S1-S4 helix to form the activated VSD dimer of Na$_v$Ab.

The homology model of the VSD dimer was embedded in the preequilibrated POPC membrane with approximately 150 mM NaCl solution. Amber ff19SB[78], Lipid21[79], TIP3P[80], and the Joung-Cheatham[81] models were employed for the Na$_v$Ab channel, the POPC lipid molecule, water, and ions, respectively. The system was composed of 1 VSD dimer, 249 POPC molecules, 21,206 water molecules, 58 Na$^+$ ions, and 68 Cl$^-$ ions. The histidine residues were assumed to be in the protonation state at pH = 7, since they face to the bulk solution. The total number of atoms in the system was 101,040. This information is tabled (Table 1).

The bad contact made in the embedding process was removed by 1000-step minimization. The MD simulation was performed for 600 ps with constant temperature (300 K) and pressure conditions (1 bar), where harmonic restraints were imposed on all atoms in the Na$_v$Ab channel with a force constant of 2 kcal/mol/Å$^2$. All MD simulations were performed using Amber22[82]. The Langevin thermostat with a collision frequency of 2 ps was used to control the temperature, and the Monte Carlo barostat with anisotropic scaling was used to control the pressure. The SHAKE algorithm[83] was used to keep the bond length having H atoms constant, enabling the use of a time step of 2 fs. The periodic

boundary condition was imposed, and long-range interactions were calculated by the particle mesh Ewald method[84] with a 10 Å cutoff in real space. Then, the MD simulation was performed for 100 ns with constant temperature (300 K) and volume conditions, where harmonic restraints were imposed on all atoms in the Na$_v$Ab channel with a force constant of 2 kcal/mol/Å$^2$. Finally, the constraints on the channels were removed, and the production run was performed for 1 μs with constant temperature (300 K) and volume conditions. The potential energy of the system did not change and α-helices were stable (Supplementary Movie 6) in the production run. 100 VSD-dimer structures every 10 ns were saved for AFM simulation, and 100 simulated AFM images were averaged to generate a simulated AFM image shown in Fig. 4c.

## Reporting summary
Further information on research design is available in the Nature Portfolio Reporting Summary linked to this article.

## Data availability
All data that support this study are available from the authors upon request. Previously published structures access from the PDB can be found using accession code 4EKW. The source data underlying Figs. 1d, 3d–f, and 4f are provided as a Source Data file. Source data are provided with this paper.

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

## Acknowledgements

A.S. acknowledges Prof. Toshio Ando, Prof. Noriyuki Kodera and Dr. Ken-ichi Umeda (Kanazawa University) for sharing instruments and operating software for HS-AFM. A.S. thanks Dr. Leonardo Puppulin and Ms. Yoko Yamamoto (Kanazawa University) for discussion and analytical assistance, respectively. We thank Dr. Damien Hall (Kanazawa University) for providing useful comments on an earlier version of this manuscript. A.S. thanks Grant-in-Aid for Scientific Research (B) (22H01919) and Challenging Research (Exploratory) (22K19290) for funding. This work was supported by the World Premier International Research Center Initiative (WPI), MEXT, Japan. K.I. acknowledge Prof. Atsunori Oshima and Mr. Yoshinori Oda (Nagoya University) for protein purification of the Na$_V$Ab KAV mutant. K.I. thanks Grants-in-Aid for Scientific Research (17K17795 and 20K09193), SEI Group CSR Foundation, Takeda Science Foundation, and Institute for Fermentation. The PCA and MD simulations were carried out on the supercomputers at the Research Center for Computational Science in Okazaki, Japan (Project: 22-IMS-C114).

## Author contributions

A.S. and K.I. designed the study. A.S. performed HS-AFM and image analysis. A.S. and M.S. optimized imaging condition of HS-AFM. M.S. supervised the project. K.I. designed and purified the NavAb construct. K.I. validated activity of the mutants by electrophysiological measurement. T.S. performed PCA, MD simulation of VSD dimer and calculation of contact probability. All authors wrote the paper.

## Competing interests

The authors declare no competing interests.
