## [Peer Review File · Nature Communications]

Voltage sensors of a Na⁺ channel dissociate from the pore domain and form inter-channel dimers in the resting stateReviewers' Comments:

Reviewer #1:

Remarks to the Author:

The manuscript by Sumino et al., suggests, based on data from high-speed atomic force microscopy experiments, that the voltage sensor domain (VSD) of the NavAB channel dissociates from the pore domain in the resting state and that VSDs from two different channels can form dimers and thereby induce cooperativity between two neighbouring ion channels

The paper is thought provoking and deviates considerably from what is generally thought of as normal gating. I find the conclusion of the paper as potentially ground-breaking. My general impression is that this is a very thorough paper, with a careful analysis, which deals with most if not all potential conflict with previously reported data. But I should also point out that I do not have direct experience with HT-AFM and thus cannot judge the methodological details connected to HTAFM.

My major comment is about how the problem is presented, the starting point of the paper. As it is written now, already in the introduction and the abstract the reader gets the impression that interactions between channels are well known and that VSD dissociating from the pore domain is something that the scientific community is discussing. In my opinion, the reports about interactions (and cooperativity) between ion channels, and that the VSD can dissociate from the pore domain are not many and not always very convincing (see examples below). The common view is that the VSD is tightly coupled to the pore domain in the resting state and that it is this interaction that closes the gate. The paper would gain from starting from a more open question: what is the resting state structure measured by other methods than x-ray or cryo-EM, and with no (apparent) restrictions of locking the channel in a resting state by disulphide bonds or similar? with no expectations of VSD dissociating from the pore domain. Cooperativity and reports on dissociated VSDs can then be dealt with in the Discussion (which already is in place).

Reference 51 proposes that there are interactions and cooperativity between ion channels. This is partly based on that Hodgkin-Huxley type equations fails to exactly predict the initiation of the action potential. First, the HH-equations are very simple and not necessarily expected to predict the exact shape of action potentials. Second, this is not a general shortcoming of the HH-equations. For instance, HH-type equations developed from voltage clamp data from the frog myelinated axon (a preparation with very high Na-channel density) almost perfectly predict experimentally recorded action potentials (Frankenhaeuser & Huxley, 1964). Thus, I think that the arguments for cooperativity are rather weak in this paper.

One previous suggestion for VSD being dislocated from the pore domain comes from the pioneering very long MD simulations by Jensen et al. (2012, ref 13 = 37). But one should keep in mind that MD simulations are not experimental data and that such dislocation can be a sign of unstable MD simulation.

One conclusion in the paper (lines 150-154) is that the S4-S5 linker was considered to be unfolded and disordered because the observed distance between the PDs and S1-S2 particles was much longer than the estimated distance. (Fig. 3I). This is based on data from a single channel (a single experiment). Is this statistically significant? If not, I suggest rephrasing it to something like "It is possible that..."

Minor points

20: Spell out what Ab stands for in NavAb

22: I suggest "surprisingly" instead of "interestingly".

30: Ionic should be ion.

31: You discuss voltage-gated cation channels (in general), but all references (2-5) are about Na channels. Either you focus on Na channels or you use more diverse references.

36-38: References are missing.

40: Initial is not a good word. There is nothing first and last if you do not refer to a specific event.

42: "Membrane potential" should be "negative membrane potential". Even 0 mV is a membrane potential.

51: "this may be the reason...". What is this referring to? Should be explained better.

51-53: This can be described better. The author is assuming that VSDs dissociate from the pore domain, but this is difficult to assume before data in the present paper is shown.

58: "in solution"? isn't the channel in membrane?

65-66: "known to be crucial for the rapid onset of action potential". This is too strong. There have been a few reports, as referenced in the paper, but that is not the same as it is known.

82-84: I suggest explaining why His-tag, and Ni²⁺-coated mica substrate was used. This is explained in lines 392-395, but for the readability it would be good to briefly mention this here also.

88: "activation starts at -100 mV". In Fig 1D it is fully open at -100 mV.

186: I suggest changing "confirming" to "suggesting".

232: "From the viewpoint of VGCC diversity, the dissociation of VSDs is also suggested." I do not understand the sentence.

234-236: To my knowledge, KvAP is a non-domain swapped channel, not a domain swapped. If I am correct, then the sentence "This indicates that VSDs can move drastically when VGCCs are functionally expressed on cells." must be removed. I do not agree that KvAP is "the most well-understood VGCC". Should be rephrased.

259: I guess that ref 35 here should be ref 37 (=ref 13)

269-271: "S4 is usually attached to PDs in the activated state, so it is natural that they dimerize when dissociated from PDs." I think this is too strong. Just because VSP and Hv1 can form dimers it is not natural that VSDs can form dimers. But it is possible. But all data together, including the new data in this paper suggests that it is "possible that the VSD itself has a dimerization ability, and dimerization is a common phenomenon in the VGCC family."

273-274: "In the case of VGCCs, it is assumed that the interaction of VSDs occurs between other channels because of the steric hindrance of PDs." I do not understand. Is "not" missing?

292: "Hodgkin-Huxley equation" should be "Hodgkin-Huxley equations".

309: I suggest changing from "the other four" to "four other" if this is what you mean.

372: I am surprised that a P/10 protocol was used. Why? This often introduces much noise. In addition, I cannot find any information about which amplifier was used and which filter was applied.

439: VDS should be VSD I guess

503: Ref 13 = ref 37

Reviewer #2:

Remarks to the Author:

The manuscript by Sumino et al. provides new and exciting results on the relationship between the voltage sensor domains and pore domains in voltage gated cation channels. The results and analysis presented are likely to be of considerable importance in the domain and in understanding the molecular basis of cellular excitability. Overall I am relatively convinced by the data and arguments of the authors though there are a number of points (in both the experiments and analysis) where perhaps some clarification could be brought.

At the experimental level there is one control that I feel is missing. This is a measurements of the degree of cleavage of the N-terminal his-tag. This is I think important to be confident about some of the measurements as the immobilization protocol relies on the C-terminal His-tag on the protein.

Structures of NavAb in different states. This section is convincing and the data well presented. Even if the relatively low number of frames for the double mutant is frustrating. In figure 1A it might help the reader to indicate the C terminal His tag explicitly.

The dynamics of the channels. This section was for me the hardest to understand. Firstly because I do not completely understand some of the details in the analysis. (i) for studying the relationship between PD and VSD (fig 3) how is the distance calculated, in particular for (a) localization of VSD where 2 particles are detected and (b) assignement of a link between VSD and PD especially in the case of the double mutant and in view of the possibility of swam/non-swap assignments. It is also clear, and remarked by the authors, that better statistics for the double mutant are desirable. (ii) in the PCA it is unclear to me how the frames were aligned, and an indication of the relative magnitudes of the different Eigen values would also be a help in understanding what these analyses (Fig3, s6, S7, S8) mean. More fundamentally I find the analysis by PCA hard to understand with the 4 fold symmetric structure where is unclear how much of the available configuration space is explored/tracked, however it is unclear to me if a different analysis (such as a collective mode analysis) would give a more accessible interpretation.

Dimerisation of the VSD - is an exciting observation. While the authors provide some clear indications I think the presentation could be a little more convincing. In particular in fig 4 (A,B,C) the use of different scales makes comparing the model 4C and the observations 4A unnecessarily difficult. A second evidence to strengthen the argument would be the observation of two PD at appropriate distances from the structures shown in fig 4A.

I am sure the authors will be able to address my concerns and clarify any points where I have misunderstood.

Response to Reviewer #1:

Reviewer comments are in italicized blue, and texts added in the revised version are quoted with revised parts in red.

The manuscript by Sumino et al., suggests, based on data from high-speed atomic force microscopy experiments, that the voltage sensor domain (VSD) of the Na_vAb channel dissociates from the pore domain in the resting state and that VSDs from two different channels can form dimers and thereby induce cooperativity between two neighbouring ion channels

The paper is thought provoking and deviates considerably from what is generally thought of as normal gating. I find the conclusion of the paper as potentially ground-breaking. My general impression is that this is a very thorough paper, with a careful analysis, which deals with most if not all potential conflict with previously reported data. But I should also point out that I do not have direct experience with HT-AFM and thus cannot judge the methodological details connected to HTAFM.

Thank you for taking your time to review our manuscript. According to your comments, we have revised the manuscript.

My major comment is about how the problem is presented, the starting point of the paper. As it is written now, already in the introduction and the abstract the reader gets the impression that interactions between channels are well known and that VSD dissociating from the pore domain is something that the scientific community is discussing. In my opinion, the reports about interactions (and cooperativity) between ion channels, and that the VSD can dissociate from the pore domain are not many and not always very convincing (see examples below). The common view is that the VSD is tightly coupled to the pore domain in the resting state and that it is this interaction that closes the gate. The paper would gain from starting from a more open question: what is the resting state structure measured by other methods than x-ray or cryo-EM, and with no (apparent) restrictions of locking the channel in a resting state by disulphide bonds or similar? with no expectations of VSD dissociating from the pore domain. Cooperativity and reports on dissociated VSDs can then be dealt with in the Discussion (which already is in place).

Thank you for your suggestion. The 2nd paragraph of the introduction was thoroughly revised to convey that the dissociation of VSD has been generally uncommon and this was first finding from the experimental side. Also, title and a subsection in the discussion “Dissociation of VSDs from PDs in the resting state” were also thoroughly revised. One sentence was added in the last paragraph to show an implication of dimerization. Revised sentences and title are the followings.

Title:

Voltage sensors of Na⁺ channels dissociate from the pore domain and form inter-channel dimers in the resting state

The 2nd paragraph of the introduction:

“The resting state structure is important among the structures obtainable during the VGCC activation cycle due to the fact that it is both the starting point of the cycle, and the structure with zero electrochemical potential. Regarding the swap-type VGCC, the VSD and PD structures of cross-linked Na_vAb in the resting state have been previously determined.⁹ On the other hand, there is little structural information about the state with a significant electrochemical potential because it is difficult to perform structural analysis under a negative membrane potential. It has only recently been reported that structure of a nonswap-type VGCC under an applied electric field.¹³ Irrespective of the presence of an electric fields, VSDs are believed to be in direct contact with PDs in all structures. However, one exception to this general rule was indicated by a molecular dynamics (MD) study in which MD simulation of the voltage-gated potassium channel predicted that VSDs dissociate from PDs when the channel closes.¹⁴ This implies that there is a possibility of VSDs generally dissociating from the PDs of VGCCs in the resting state. However, this postulated dissociated structure of VSD has not yet been demonstrated experimentally, and thus the question arises as to whether or not this is a computational artifact. If the dissociation event is indeed proved to be real, it would be important to reveal the role of such dissociation in the activation cycle.” (Line 40-55)

The subsection “Dissociation of VSDs from PDs in the resting state”:

“The VSDs have been considered to be tightly bound to the PDs in the WT so far. This tight bound structure corresponds to a fully activated structure in the depolarization state in this study. In the E32Q/N49K mutant, in which some VSDs were not fully activated without a membrane potential, all VSDs were found to dissociate from

the PDs. The N49K mutant, whose voltage dependency is **between the WT and E32Q/N49K**, appeared to have VSDs that were tightly attached to the PDs, as in the WT, but the PCA revealed that the interaction between VSDs and PDs was diminished (Figure 3J, K and L). Elinder et al. reported that the distance between the extracellular tops of the S4 and S5 helices was 8 Å in the open state, **but was elongated** > 20 Å in the closed state.³⁶ **The present observations are consistent with this result. During hundreds μ s MD simulation of the K_v channel, a channel of the VGCC family, it was observed that the S4-S5 linker unraveled and the VSDs detached from the PDs and moved far away.¹⁴ Our results offer experimental evidence for this MD simulation result. Even in the N49K crystal structure, VSDs have been observed to be slightly farther from the PDs compared to those in the WT crystal structure.⁴ It is expected that the tight binding of VSDs to PDs is strengthened by structural changes in the VSDs associated with the depolarization of the membrane potential. Accordingly, a consideration of the association and dissociation of VSDs from PDs would be beneficial for elucidating the gating mechanism of VGCCs.**

The single-particle structure determined for the resting state of the swap-type VGCC was derived from a mutant channel in which two residues were replaced by cysteine to form a disulfide bond between VSD and PD.⁹ Without the disulfide bond, the VSD structure of this channel formed the activated state,⁹ and has not therefore been determinable by single-particle analysis. Therefore, the VGCC structure in the resting state without the disulfide bond has not been reported. This implies that the known structure in the resting state was one of the snapshots fixed by the disulfide bond and that VSDs may undergo more significant movement when VGCCs are functionally expressed without the disulfide bond in the resting state. Furthermore, even for the resting state channel structure determined with the disulfide bond present, the cytosolic-side conformation of the S4 helix differed from that of the activated state because the extracellular side of VSDs and PDs were fixed by the disulfide bond.⁹ Therefore, under natural conditions without the disulfide bond, the possibility is extant that the cytosolic-side of the S4 helix has other structures, such as the elongated S4-S5 linker.

In this study, we performed HS-AFM on Na_vAb channels. Use of HS-AFM may prove to be the key to observing the VSDs dissociation from PDs in the resting state. Structures of dissociated VSDs have not been observed thus far because it is difficult to detect them with single-particle-based structural analysis, which requires averaging of a great many number of molecular structures.^{9,38} Even if VSD dissociates in a natural membrane, it will be difficult to classify and determine the dissociated structure of VSD by image classification and averaging of single particle analysis, because the structures are various (Figure 2). Also, various structures might be consolidated into a single

conformation in the process of crystal growth. In stark contrast to such considerations, HS-AFM is a method that can obtain structural information from a single molecule, which allowed us to observe the dissociated VSD structure.

VGCCs include a swap-type VGCC in which the VSD contacts the PD of the adjacent subunit and a nonswap-type VGCC in which the VSD contacts the PD of the same subunit.⁶ Comparing these two types, we note that the position of the VSD is very different. Even in swap-type VGCCs, there are various VSD positions.³⁹ Thus, there is significant diversity in the VSD positions, which implies that the VSD positions have changed during the molecular evolution of VGCCs. It suggests that VSD could significantly move around the PD. In a recent study, the VSD arrangement of K_vAP channel, the first VGCC whose structure was determined,⁴⁰ was finally confirmed to be a nonswap type after a period of controversy.³⁸ This further suggests that it is difficult to determine the actual position of VSD, and it is necessary to analyze VSD arrangement with various methods. Therefore, it will be interesting to obtain more information about the possible dissociation of VSD using HS-AFM.” (Line 222-272)

The last paragraph:

“Therefore, abnormality in inter-channel association in the VGCC family, such as mutations in the residues between VSDs, would be the cause of some channelopathies.” (Line 344-346)

Reference 51 proposes that there are interactions and cooperativity between ion channels. This is partly based on that Hodgkin-Huxley type equations fails to exactly predict the initiation of the action potential. First, the HH-equations are very simple and not necessarily expected to predict the exact shape of action potentials. Second, this is not a general shortcoming of the HH-equations. For instance, HH-type equations developed from voltage clamp data from the frog myelinated axon (a preparation with very high Na-channel density) almost perfectly predict experimentally recorded action potentials (Frankenhaeuser & Huxley, 1964). Thus, I think that the arguments for cooperativity are rather weak in this paper.

Discussion related ref. 51 (now 53) was revised to note that there is a controversy and to slightly tone down our claim. “strong” in the last sentence in the same paragraph was deleted. Revised sentences are the followings.

“Historically, the Hodgkin-Huxley equations **have** been used to model action potential generation.^{51,52} Naundorf et al. have **proposed** that the cooperative activation of sodium channels is important for reproducing the experimental dynamics of action potential initiation,⁵³ **though** the molecular basis of functional cooperativity is unclear. **There is still a controversy over the cooperativity,**^{54,55} **while** the functional cooperativity of Na⁺ channels has been extensively investigated.^{50,53,56–63} Since it has been reported that dimers of H_v channels are cooperatively activated³⁰, the dimerization of VSDs **might be involved in** the cooperative activation of Na_v channels.” (Line 320-327)

Last sentence:

“Integrating all this information, we expect that the dimerization of dissociated VSDs stabilizes the resting state of channels and is a **strong** candidate for the molecular basis of the cooperative activation of Na_v channels (Figure 5).” (Line 339-341)

One previous suggestion for VSD being dislocated from the pore domain comes from the pioneering very long MD simulations by Jensen et al. (2012, ref 13 = 37). But one should keep in mind that MD simulations are not experimental data and that such dislocation can be a sign of unstable MD simulation.

Accordingly, the 2nd paragraph of the introduction was revised.

One conclusion in the paper (lines 150-154) is that the S4-S5 linker was considered to be unfolded and disordered because the observed distance between the PDs and S1-S2 particles was much longer than the estimated distance. (Fig. 3I). This is based on data from a single channel (a single experiment). Is this statistically significant? If not, I suggest rephrasing it to something like “It is possible that...”

We corrected it accordingly.

Minor points

20: Spell out what Ab stands for in NavAb

We corrected it accordingly.

22: I suggest “surprisingly” instead of “interestingly”.

We corrected it accordingly.

30: Ionic should be ion.

We corrected it accordingly.

31: You discuss voltage-gated cation channels (in general), but all references (2-5) are about Na channels. Either you focus on Na channels or you use more diverse references.

We cited the papers about potassium and calcium channels.

36-38: References are missing.

We cited ref. 12 there.

40: Initial is not a good word. There is nothing first and last if you do no refer to a specific event.

The sentence was revised to specify that the structure is not under electrochemical potential. Revised sentence is in the 2nd paragraph of the introduction, shown previously.

42: “Membrane potential” should be “negative membrane potential”. Even 0 mV is a membrane potential.

We agree that it was vague and corrected it accordingly.

51: *“this may be the reason...”. What is this referring to? Should be explained better.*

The sentence was revised as follows.

“For collecting and averaging similar structures, cross-linking is sometimes necessary for such structural analysis.”⁹ (Line 59-60)

51-53: *This can be described better. The author is assuming that VSDs dissociates from the pore domain, but this is difficult to assume before data in the present paper is shown.*

Thank you for your comment. The sentence was revised as follows.

“If VSDs really move freely away from PDs as found in the MD simulation, then the averaging performed in single-particle analysis is not suitable.” (Line 59-61)

58: *“in solution”? isn't the channel in membrane?*

The channel was in membrane, not in solution. So, “in solution” was deleted.

65-66: *“known to be crucial for the rapid onset of action potential”. This is too strong. There have been a few reports, as referenced in the paper, but that is not the same as it is known.*

We agree that it was too strong, so deleted it.

82-84: *I suggest explaining why His-tag, and Ni²⁺-coated mica substrate was used. This is explained in lines 392-395, but for the readability it would be good to briefly mention this here also.*

Thank you for the comment. We added “of the PD to suppress diffusion in the bilayer” as follows.

“To reliably observe the single-molecule structural dynamics of the NavAb channel by HS-AFM, we truncated the C-terminal cytoplasmic domain (230-268) and added a His-tag on the C-terminus **of the PD to suppress diffusion in the bilayer** (for readability, this paper will omit describing the notation of the deletion of the C-terminus and the addition of a His-tag).” (Line 87-91)

88: “activation starts at -100 mV”. In Fig 1D it is fully open at -100 mV.

We are sorry, it was corrected as -120 mV.

186: I suggest changing “confirming” to “suggesting”.

We corrected it accordingly.

232: “From the viewpoint of VGCC diversity, the dissociation of VSDs is also suggested.” I do not understand the sentence.

Following your suggestion, we rewrote the section “Dissociation of VSDs from PDs in the resting state,” and this sentence was deleted.

234-236: To my knowledge, KvAP is a non-domain swapped channel, not a domain swapped. If I am correct, then the sentence “This indicates that VSDs can move drastically when VGCCs are functionally expressed on cells.” must be removed. I do not agree that KvAP is “the most well-understood VGCC”. Should be rephrased.

I understood that first structure of KvAP, known as puddle model, is swap-type, because predicted model of up-conformation of puddle VSD is swapped. However, the KvAP structure was recently revealed by electron microscopy as nonswap. Therefore, it is correct to say that it was recently confirmed to be nonswap rather than a channel that can be either swap or nonwap, as you pointed out. Thus, we rewrote to the following sentence.

“In a recent study, the VSD arrangement of K_vAP channel, the first VGCC whose structure was determined,⁴⁰ was finally confirmed to be a nonswap type after a period of controversy.³⁸” (Line 267-269)

259: *I guess that ref 35 here should be ref 37 (=ref 13)*

Thank you for pointing this out. We have corrected the reference.

269-271: *“S4 is usually attached to PDs in the activated state, so it is natural that they dimerize when dissociated from PDs.” I think this is too strong. Just because VSP and Hv1 can form dimers it is not natural that VSDs can form dimers. But it is possible. But all data together, including the new data in this paper suggests that it is “possible that the VSD itself has a dimerization ability, and dimerization is a common phenomenon in the VGCC family.”*

We agree with your point. So “so it is natural that they dimerize when dissociated from PDs” was removed, and the next sentence was revised according to your suggestion.

“S4 is usually attached to PDs in the activated state. Thus, it **is** possible that the VSD itself has a dimerization ability, and dimerization is a common phenomenon in the VGCC family.” (Line 297-299)

273-274: *“In the case of VGCCs, it is assumed that the interaction of VSDs occurs between other channels because of the steric hindrance of PDs.” I do not understand. Is “not” missing?*

We are sorry, “not” was missed. Also, the sentence was vague so it was revised as follows.

“In the case of VGCCs, it **has not been** assumed that **there is** interaction **between** VSDs **within a channel** because of the steric hindrance of PDs **since VSDs were considered to be in direct contact with PDs.**” (Line 300-302)

292: “Hodgkin-Huxley equation” should be “Hodgkin-Huxley equations”.

We corrected it accordingly.

309: I suggest changing from “the other four” to “four other” if this is what you mean.

We corrected it accordingly.

372: I am surprised that a P/10 protocol was used. Why? This often introduces much noise. In addition, I cannot find any information about which amplifier was used and which filter was applied.

We understand your noise concerns. The figure below is a representative current trace of the tail-current activation used to create Fig1d. As you can see, we can measure the tail current from the data. We will add this data to the supplementary information if necessary.

And we added the model number of the amplifier and the application information used for measurement were also added as follows.

“All experiments were conducted at $25 \pm 2^\circ\text{C}$ using a whole cell patch clamp recording mode with a HEKA EPC 10 amplifier and Patch master data acquisition software (v2x73). Data export was done using IGOR 6.37 and NeuroMatic (version 3.0b).⁷³” (Line 408-410)

439: VDS should be VSD I guess

We corrected it accordingly.

503: Ref 13 = ref 37

Thank you for pointing this out. We have corrected the duplication.

Response to Reviewer #2:

Reviewer comments are in italicized blue, and texts added in the revised version are quoted with revised parts in red.

The manuscript by Sumino et al. provides new and exciting results on the relationship between the voltage sensor domains and pore domains in voltage gated cation channels. The results and analysis presented are likely to be of considerable importance in the domain and in understanding the molecular basis of cellular excitability. Overall I am relatively convinced by the data and arguments of the authors though there are a number of points (in both the experiments and analysis) where perhaps some clarification could be brought.

Thank you for taking your time to review our manuscript. According to your comments, we have revised the manuscript.

At the experimental level there is one control that I feel is missing. This is a measurements of the degree of cleavage of the N-terminal his-tag. This is I think important to be confident about some of the measurements as the immobilization protocol relies on the C-terminal His-tag on the protein.

Following suggestion, we modified the method. We added that we confirmed in the previous paper that N-terminal His-tag was completely digested.

“After washing, the protein was eluted with 300 mM imidazole, and the N-terminal His-tag was **completely** removed by **5U/ml** thrombin (Fujifilm Wako; 206-18411) digestion **in eluted solution added 2mM CaCl₂** (overnight, 4°C). **This digestion condition was established in our previous study.⁴**” (Line 376-379)

Structures of NavAb in different states. This section is convincing and the data well presented. Even if the relatively low number of frames for the double mutant is frustrating. In figure 1A it might help the reader to indicate the C terminal His tag explicitly.

Thank you for the comment. Fig. 1a is intended to illustrate the general structure of NavAb, and adding the His-tag to Fig. 1a would change the intent, so instead we have

changed the color and added “on PD” to make the position of His-tag in Fig. 1c stand out. Also, to clearly describe the position of His-tag, we added description “of the PD” in the text as follows.

“we truncated the C-terminal cytoplasmic domain (230-268) and added a His-tag on the C-terminus **of the PD to suppress diffusion in the bilayer** (for readability, this paper will omit describing the notation of the deletion of the C-terminus and the addition of a His-tag).” (Line 88-91)

The dynamics of the channels. This section was for me the hardest to understand. Firstly because I do not completely understand some of the details in the analysis. (i) for studying the relationship between PD and VSD (fig 3) how is the distance calculated, in particular for (a) localization of VSD where 2 particles are detected and (b) assignment of a link between VSD and PD especially in the case of the double mutant and in view of the possibility of swap/non-swap assignments. It is also clear, and remarked by the authors, that better statistics for the double mutant are desirable. (ii) in the PCA it is unclear to me how the frames were aligned, and an indication of the relative magnitudes of the different Eigen values would also be a help in understanding what these analyses (Fig3, s6, S7, S8) mean. More fundamentally I find the analysis by PCA hard to understand with the 4 fold symmetric structure where it is unclear how much of the available configuration space is explored/tracked, however it is unclear to me if a different analysis (such as a collective mode analysis) would give a more accessible interpretation.

Thank you for comments.

We used center-to-center distance for this analysis, and clarified the definition of the distance in the text. For particle assignment, we cannot visualize the linker between PD and VSD since the linker is behind the bilayer. As described in Figure 1B, Na_vAb has domain-swapped structure. Considering this structure, we assigned VSDs as illustrated in Figure 2 D, H and O. As you pointed out, the most confusing case is the double mutant. We understand some statistical analysis may work, but here we wanted to estimate the minimum distance of PD-VSD pair, so we simply assumed that the closest VSD (pair of two particles derived from S1-S2 and S3-S4) in the clockwise direction from the PD as the VSD linked to the PD and that the particle closer to PD in the VSD was assigned as

S3-S4, and the particle farther away was assigned as S1-S2. To clearly describe this, the following sentences were added.

“The pair of VSDs and PDs were assigned by taking the swapped structure into account (Figures 2D, H, and O), and the PD–VSD distances were estimated. The distance between center of PD particle and center of VSD particle was defined as the PD–VSD distance. In the E32Q/N49K mutant, we assumed the closest VSD in the clockwise direction from the PD as the VSD linked to the PD.” (Line 151-155)

For PCA calculation, alignment was not performed since the drift of PDs was small. It was written as follows.

“Alignment of the frames was not performed, since the drift of PDs was small.” (Line 467-468)

In our understanding, PCA is the same as collective mode you suggested (see Kitao & Go, *Curr. Opin. Struct. Biol.* 9, 164 (1999), He et al., *Biophys. J.* 100, 1058 (2011).), so the below sentence was added.

“PCA can visualize the so-called collective mode.” (Line 169)

For eigenvalues, the percentage in Figs. S6-8 were based on eigenvalues, but we are sorry that explanation was not enough. Such information was lack in the main figure (Fig. 3J, K, L), so it was added. The revised sentences are follows.

“Below the eigenvectors, the ratio of the eigenvalues of the relevant mode to the sum of the eigenvalues of all modes is shown as a percentage.” (Captions in Figures 3, S6-8)

“By diagonalizing C , eigenvectors and eigenvalues were computed.” (Line 470-471)

When revising figures, we found that parts of eigenvectors were missing in Figs. 3L and S8 mol 2, so they were updated.

Four-fold symmetry was not considered in PCA, because some particles, especially VSD particles, did not show symmetric motion. Note that, even though four-fold symmetry

was not taken into account, symmetric rotational mode of PDs was often visualized. Added sentences are follows.

“In this analysis, four-fold symmetry was not taken into account to consider asymmetric motion of VSDs and PDs. In other words, each particle was considered to move independently.” (Line 468-470)

Dimerisation of the VSD - is an exciting observation. While the authors provide some clear indications I think the presentation could be a little more convincing. In particular in fig 4 (A,B,C) the use of different scales makes comparing the model 4C and the observations 4A unnecessarily difficult. A second evidence to strengthen the argument would be the observation of two PD at appropriate distances from the structures shown in fig 4A.

Thank you for comments. We modified the scales of Fig. 4A, B, and C to be close (the scales of B and C are same). The length of the scale bar in Fig. 4A was modified from 5 nm to 3 nm to match to those in B and C. Additionally, we added overlay image of B and C as D. One caution should be the orientation of molecule in Figure B and C (or D), which was written as follows.

“Note that the molecular orientation in panel B is the same as that in panel C.”
(Caption in Figure 4D)

For the latter comment, as shown in Fig. 2L, VSDs rapidly move in lipid membranes, so it was necessary to observe with a fast scan speed of about 0.2 sec/frame to clearly image VSD particles. However, scanning a wide range where two NavAbs exist (e.g., 60 nm x 60 nm area) at this scan speed caused significant damages to the sample, making it difficult to measure the double mutant, which has an unstable structure. In this study, due to the problem of the combination of sample stability and the scanning conditions described above, we were unable to obtain images of multiple resting channels connected to each other. In the future, we would like to image cross-links between resting channels by controlling the membrane potential with a more stable WT channel.

I am sure the authors will be able to address my concerns and clarify any points where I have misunderstood.

We hope our replies addressed all your concerns.

Reviewers' Comments:

Reviewer #1:

Remarks to the Author:

Most of my comments have been dealt with in a satisfactory way. However, I still have a few comments. (One problem was that the manuscript file in the revised version was reproduced four times. In some of the versions there were larger sections in red, in some of the versions there were just a few red sections. I have focused on the second variant where there were several red sections. I hope this was the correct one.)

TITLE: The new title is better, but I suggest to alter "Na⁺ channels" to "a Na⁺ channel" because you only have data for one channel (NavAb).

COOPERATIVITY: Early in the abstract (lines 19-20) it is said that "Another problem is that the key structure facilitating positive cooperativity in the rising phase of the action potential is unknown". I think it is inappropriate to use this as an argument to undertake the present investigation. Cooperativity is in fact not at all mentioned in the much longer Introduction. It appears first in the Discussion (which I think is fine). I suggest removing this sentence. Possibly a sentence with comments about cooperativity can be added at the latter part of the abstract (corresponding to the Discussion about cooperativity).

OTHER COMMENTS

Lines 40-42: The new sentence "The resting state structure is important among the structures obtainable during the VGCC activation cycle due to the fact that it is both the starting point of the cycle, and the structure with zero electrochemical potential" is unclear. What does "obtainable" mean? Are other structures not possible to obtain? A cycle does not have a starting point; it is a cycle. What is meant by zero electrochemical potential? I suggest to rewrite it.

Lines 43-45: The partly rewritten sentence "On the other hand, there is little structural information about the state with a significant electrochemical potential because it is difficult to perform structural analysis under a negative membrane potential" is not clear. What is meant by "significant electrochemical potential"? I guess you mean that we have little structural information of ion channels at normal resting potential (-70 mV). I also think that "zero" electrochemical potential in my previous comment now has become "significant". I suggest to spell out what is meant.

Lines 267-269: "In a recent study, the VSD arrangement of KvAP channel, the first VGCC whose structure was determined,40 was finally confirmed to be a nonswap type after a period of controversy.38". I do not know which controversy this is referred to. Reference 38 does not mention KvAP. Please clarify.

Reviewer #2:

Remarks to the Author:

The authors have satisfactorily answered all of the concerns that I expressed in my initial review.

Reviewer #3:

Remarks to the Author:

This reviewer was asked to evaluate the technical aspects of using high-speed AFM. The authors stated that, "Use of HS-AFM may prove to be the key to observing the VSDs dissociation from PDs in the resting state." Implication is that HS-AFM generated data and images that were critical to drawing the conclusions of the study. HS-AFM has been known for some time and the authors referenced the

2012 Nature paper without providing experimental details. Instead, a custom-built HS-AFM was mentioned. Use of HS-AFM can depend on the sampling rate and potentially the rate at which internal reorganization can occur, such as the rate of dissociation. While the authors appear to have sufficient knowledge and understanding of HS-AFM, it would be helpful to include additional details of the custom-built HS-AFM to convince the reader that proper precautions have been considered and implemented to avoid potential pitfalls. For example, typical AFM images of $\sim 10 \times 10$ nm (Fig. 2). What was the size of AFM tips and their geometrical shape along with the sampling rate? What criteria were applied to optimize AFM imaging? Interestingly, the time scale of the images shown in Figure 2 was ~ 20 s. One would presume voltage-gated ion channels operate on a much short time scale.

Reviewer #1 (Remarks to the Author):

Thanks for the careful reading and kind suggestions. The comments we received allowed us to rewrite the content clearer.

Most of my comments have been dealt with in a satisfactorily way. However, I still have a few comments. (One problem was that the manuscript file in the revised version was reproduced four times. In some of the versions there were larger sections in red, in some of the versions there were just a few red sections. I have focused on the second variant where there were several red sections. I hope this was the correct one.)

We are sorry for confusing you. The latest version is the second variant as the reviewer commented. Our revision was also based on that version.

TITLE: The new title is better, but I suggest to alter “Na⁺ channels” to “a Na⁺ channel” because you only have data for one channel (NavAb).

Following the reviewer’s suggestion, title was changed to “Voltage sensors of a Na⁺ channel dissociate from the pore domain and form inter-channel dimers in the resting state”

COOPERATIVITY: Early in the abstract (lines 19-20) it is said that “Another problem is that the key structure facilitating positive cooperativity in the rising phase of the action potential is unknown”.

I think it is inappropriate to use this as an argument to undertake the present investigation. Cooperativity is in fact not at all mentioned in the much longer Introduction. It appears first in the Discussion (which I think is fine). I suggest removing this sentence. Possibly a sentence with comments about cooperativity can be added at the latter part of the abstract (corresponding to the Discussion about cooperativity).

Following the reviewer’s advice, we removed the Line19-20 part and added some words to the last sentence of the abstract as follows:

This dimerization would occur at a realistic channel density, offering a potential explanation for the facilitation of positive cooperativity of channel activity in the rising phase of the action potential.

OTHER COMMENTS

Lines 40-42: The new sentence “The resting state structure is important among the structures obtainable during the VGCC activation cycle due to the fact that it is both the starting point of the cycle, and the structure with zero electrochemical potential” is unclear. What does “obtainable” mean? Are other structures not possible to obtain? A cycle does not have a starting point; it is a cycle. What is meant by zero electrochemical potential? I suggest to rewrite it.

Lines 43-45: The partly rewritten sentence “On the other hand, there is little structural information about the state with a significant electrochemical potential because it is difficult to perform structural analysis under a negative membrane potential” is not clear. What is meant by “significant electrochemical potential”? I guess you mean that we have little structural information of ion channels at normal resting potential (-70 mV). I also think that “zero” electrochemical potential in my previous comment now has become “significant”. I suggest to spell out what is meant.

We removed Lines 40-42 part and changed the Line43-45 part as follows:
“The resting state structure is the least understood structure in the VGCC activation cycle. This is because the formation of this structure requires the resting membrane potential, but it is difficult to analyze the structure at resting membrane potential.”

And the following two sentences were replaced:

“It has only recently been reported that structure of a nonswap-type VGCC under an applied electric field.¹³ “

and

“Regarding the swap-type VGCC, the VSD and PD structures of cross-linked Na_vAb in the resting state have been previously determined.⁹”

Additionally, we modified following description at line 43.

“In all the determined structures including those in the activated state,” which was originally “Irrespective of the presence of an electric fields,”

Lines 267-269: “In a recent study, the VSD arrangement of KvAP channel, the

first VGCC whose structure was determined,40 was finally confirmed to be a nonswap type after a period of controversy.38". I do not know which controversy this is referred to. Reference 38 does not mention KvAP. Please clarify.

Sorry, reference number in this part was 37 instead of 38.

Reviewer #2 (Remarks to the Author):

The authors have satisfactorily answered all of the concerns that I expressed in my initial review.

Thank you again for your comments in your initial review.

Reviewer #3 (Remarks to the Author):

HS-AFM has been known for some time and the authors referenced the 2012 Nature paper without providing experimental details. Instead, a custom-built HS-AFM was mentioned. Use of HS-AFM can depend on the sampling rate and potentially the rate at which internal reorganization can occur, such as the rate of dissociation. While the authors appear to have sufficient knowledge and understanding of HS-AFM, it would be helpful to include additional details of the custom-built HS-AFM to convince the reader that proper precautions have been considered and implemented to avoid potential pitfalls. For example, typical AFM images of ~ 10 x 10 nm (Fig. 2). What was the size of AFM tips and their geometrical shape along with the sampling rate? What criteria were applied to optimize AFM imaging? Interestingly, the time scale of the images shown in Figure 2 was ~ 20 s. One would presume voltage-gated ion channels operate on a much short time scale.

Thank you for your comments. We added the experimental details about HS-AFM as follows:

“The shape of the tip was conical, and the tip radius was less than 2 nm based on the obtained resolution.” (Line 437-438)

“Scanning rates were 5 or 10 frames/sec (the actual rate is described in the figure

caption). Typical scanning range was 50 nm × 30 nm. These scanning conditions were optimized to obtain high-resolution images without disrupting the tetrameric structure of channels. Any faster scanning rates or larger scanning area would destroy the structure of the sample.” (Line 442-446)

Although we used laboratory-build HS-AFM, its basic performance is comparable to that of commercially available products, and we did not add any special features.

As you mentioned, ion channels respond to change in membrane potential in milli seconds, and this is much faster than the scanning speed of HS-AFM observation. The significance of the high-speed AFM observations in this paper was to clearly visualize the structure of the channel with fluctuations on a sub-second scale in the steady state, and to evaluate the strength of the interdomain interaction by analyzing the fluctuations. In the future, it would be wonderful if the time resolution of high-speed AFM could be dramatically improved so that the process of voltage-dependent structural changes could be visualized.

Reviewers' Comments:

Reviewer #3:

Remarks to the Author:

I have no other comments to add.